# Cellular and Subcellular Characteristics of Neuromuscular Junctions in Muscles with Disparate Duty Cycles and Myofiber Profiles

**DOI:** 10.3390/cells12030361

**Published:** 2023-01-18

**Authors:** Michael R. Deschenes, Mia K. Mifsud, Leah G. Patek, Rachel E. Flannery

**Affiliations:** 1Department of Kinesiology & Health Sciences, College of William & Mary, Williamsburg, VA 23187, USA; 2Program in Neuroscience, College of William & Mary, Williamsburg, VA 23187, USA

**Keywords:** synapse, NMJ, endplate, acetylcholine, myofiber

## Abstract

The neuromuscular system accounts for a large portion (~40%) of whole body mass while enabling body movement, including physical work and exercise. At the core of this system is the neuromuscular junction (NMJ) which is the vital synapse transducing electrical impulses from the motor neurons to their post-synaptic myofibers. Recent findings suggest that subcellular features (active zones) of the NMJ are distinctly sensitive to changes in activity relative to cellular features (nerve terminal branches, vesicles, receptors) of the NMJ. In the present investigation, muscles with different recruitment patterns, functions, and myofiber type profiles (soleus, plantaris, extensor digitorum longus [EDL]) were studied to quantify both cellular and subcellular NMJ characteristics along with myofiber type profiles. Results indicated that, in general, dimensions of subcellular components of NMJs mirrored cellular NMJ features when examining inter-muscle NMJ architecture. Typically, it was noted that the NMJs of the soleus, with its most pronounced recruitment pattern, were larger (*p* < 0.05) than NMJs of less recruited muscles. Moreover, it was revealed that myofiber size did not dictate NMJ size as soleus muscles displayed the smallest fibers (*p* < 0.05) while the plantaris muscles exhibited the largest fibers. In total, these data show that activity determines the size of NMJs and that generally, size dimensions of cellular and subcellular components of the NMJ are matched, and that the size of NMJs and their underlying myofibers are uncoupled.

## 1. Introduction

Proper structure and function of the neuromuscular system are required to conduct both essential daily functional activities as well as recreational activities that add pleasure to our lives. It is known that for an effective neuromuscular function to occur, there must be proper communication between motor neurons and the myofibers they innervate. This communication takes place at the neuromuscular junction (NMJ), which is the vital synapse joining pre-synaptic motor neurons with their targeted post-synaptic myofibers.

These NMJs exhibit specific cellular components such as the number and length of pre-synaptic nerve terminal branches and the number of neurotransmitter-containing vesicles, along with post-synaptic measures such as the area of and perimeter length around motor endplates expressing receptors for the neurotransmitter. In addition, the same NMJs display distinct subcellular features, particularly pre-synaptic active zones, which serve to anchor or “dock” pre-synaptic vesicles containing acetylcholine (ACh) in their proper positions of apposition across from post-synaptic ACh receptors located at the endplate region of the sarcolemma. Importantly, research has revealed that the morphological characteristics of these active zones are critical to the efficiency of the NMJ, even to the extent that active zone size is a better indicator of ACh release upon electrical impulse than the total size of the NMJ [1]. Moreover, it has been demonstrated that alterations in the size of active zones act as forerunners to long-term modifications in neuromuscular transmission [2]. 

Recently, it has been reported that changes in neuromuscular activity stimulate the re-modeling of active zones at pre-synaptic motor nerve terminals. More specifically, increased activity led to increased active zone size [3], while decreased activity resulted in reduced active zone dimensions [4]. Results from those two findings suggest that the active zone or subcellular aspects of the NMJ, were more sensitive to alterations in activity level than were the cellular features more commonly examined. It must be taken into account, however, that in those studies, differences in neuromuscular activity were imposed by having small animals participate in a managed program with run training as a model of increased activity or by forced participation in a muscle unloading protocol to impose subtotal muscle disuse. In light of this, perhaps the effects of differences in neuromuscular activity would be more clearly revealed by investigating both cellular and subcellular morphological characteristics of NMJs in muscles with disparate patterns of natural recruitment and activation of the neuromuscular system. With this in mind, the present investigation was carried out to determine if the differing duty cycles or neuromuscular recruitment evident in soleus, plantaris, and extensor digitorum longus muscles were associated with similar or different morphological characteristics in pre-synaptic nerve terminals (including active zones), post-synaptic endplates, and myofiber profiles, i.e., myocyte size and type, in those three muscles.

## 2. Experimental Procedures

### 2.1. Animal Care and Muscle Selection

Ten adult (9-month-old) male Wistar rats purchased from Charles River Laboratories (Wilmington, MA, USA) were included in this study. Animals were euthanized following anesthesia via Ketamine/Xylazine cocktail injection. The soleus, plantaris, and EDL muscles of the lower leg were then dissected out, cleared of fat and connective tissue, and frozen at approximate resting length in isopentane chilled with dry ice. Muscles were then stored at −85 °C until analysis. These muscles were selected for study because it has been shown that each has a unique function and recruitment pattern [5,6,7], along with a different myofiber type profile [8]. For example, the soleus is primarily a slow-twitch muscle that has a high-duty cycle due to its role as the main postural muscle and displays a high percentage of Type I myofibers. The plantaris, while similar to the soleus in that it serves as an ankle extensor, is mainly composed of Type II myofibers and, although playing an important role in locomotor activity, is not considered a postural muscle, thus exhibiting a lower duty cycle than the soleus. Finally, the EDL muscle differs from both the soleus and the plantaris in that it is an ankle flexor, not an extensor, and is not significantly involved in locomotion, but, similar to the plantaris, it is primarily composed of Type II myofibers. As a result, investigating these muscles provided valuable insight into different muscles with varying functions, recruitment patterns, and myofiber compositions, thus enabling a comprehensive view of the effects of recruitment, fiber type, and function on cellular and subcellular features of the NMJ. All treatment and care procedures employed in this investigation were approved by the institution’s animal care and use committee, which operates in full compliance with the National Institutes of Health Guide for the Care and Use of Laboratory Animals as revised in 2011. Throughout experimentation, all efforts were made to minimize the number of animals used and their suffering.

### 2.2. Cytofluorescent Staining of Cellular Components of the NMJ

In order to visualize gross or cellular aspects of NMJs, 50 µm thick longitudinal sections from the middle one-third of the whole muscle were initially obtained at −20 °C on a cryostat (Cryocut 1800, Reichrt-Jung, Nubloch, Germany). In order to prevent contraction, microscope slides were pre-treated in a 3% EDTA solution as previously described [9]. Sections were then washed 4 × 15 min in phosphate-buffered saline (PBS) with 1% bovine serum albumin (BSA). Muscle sections were then incubated in a humidified chamber overnight at 4 °C in the supernatant of RT-97 antibody (Developmental Studies Hybridoma Bank, University of Iowa) diluted 1:20 in PBS with 1% BSA. The RT-97 antibody reacts with non-myelinated segments of pre-synaptic nerve terminals [10]. The next day, sections were washed 4 × 15 min in PBS with 1% BSA before being incubated for 2 h at room temperature in AlexaFluor 488 conjugated secondary antibody (Thermo Fisher Scientific, Ashville, NC, USA) diluted 1:200 in PBS with 1% BSA. Sections were then washed 4 × 15 min in PBS with 1% BSA before being incubated overnight in a humidified chamber at 4 °C in a solution containing rhodamine-conjugated α-Bungarotoxin (BTX; Thermo Fisher Scientific, Ashville, NC, USA) diluted 1:500 in PBS with 1% BSA, along with anti-synaptophysin (MP Biomedicals, Solon, OH, USA) diluted 1:150 in PBS with 1% BSA. BTX recognizes post-synaptic ACh receptors, while anti-synaptophysin reacts with pre-synaptic vesicles containing ACh. Indeed, synaptophysin is the most abundant protein found in the pre-synaptic vesicular membrane [11]. The following day, sections were washed 4 × 15 min in PBS with 1% BSA before they were incubated for 2 h in a humidified chamber at room temperature in AlexaFluor 647 (Thermo Fisher Scientific, Ashville, NC, USA) diluted 1:200 in PBS with 1% BSA to visualize ACh containing pre-synaptic vesicles. Sections were again washed 4 × 15 min in PBS with 1% BSA before being lightly coated with ProLong Gold (Thermo Fisher Scientific, Ashville, NC, USA) and having coverslips applied. Representative images of the cellular features of NMJs are found in Figure 1. Manually and automatically drawn lines around post-synaptic endplates are displayed in Figure 2. The manually drawn lines were used to quantify variables referred to as “Total Perimeter” and “Total Area.” Lines drawn automatically using the Image-Pro Plus software were used to measure morphological variables termed “Stained Perimeter” and “Stained Area,” i.e., only areas actually stained, and not unstained regions interspersed within the overall structure.

In order to identify subcellular aspects of NMJs on those same muscles, additional longitudinal sections from the same muscles were collected, but at a thickness of 20 µm. A different staining procedure was used on those sections in order to identify active zones and post-synaptic ACh receptors. Specifically, sections were incubated in 4% paraformaldehyde at room temperature for 10 min. Then sections were washed 4 × 15 min in PBS with 1% BSA before being incubated in 0.25% Triton X-100 (Sigma Aldrich, St. Louis, MO, USA) for 5 min at room temperature. Following this, slides were incubated in 3% BSA at room temperature for 30 min. Sections were then incubated in a cocktail of anti-Bassoon (Novus Biochemicals, Centennial, CO, USA) and anti-calcium P/Q channel (Synaptic Systems, Goettingen, Germany), both diluted 1:200 in PBS with 1% BSA overnight in a humidified chamber at room temperature. The following day, sections were washed 4 × 15 min in PBS with 1% BSA. The sections were then incubated with rhodamine-conjugated BTX diluted 1:500. AlexaFluor 488 (Thermo Fisher Scientific, Ashville, NC, USA) secondary antibody for Bassoon visualization (Bassoon is a scaffolding protein that is a main structural constituent of the active zone), and AlexaFluor 647 secondary antibody for P/Q calcium channel visualization (Thermo Fisher Scientific, Ashville, NC, USA) were then applied to muscle sections. Both secondary antibodies were diluted 1:150 in PBS with 1% BSA before being incubated for 90 min at room temperature. Following this staining incubation, sections were washed for 3 × 5 min in PBS with 1% BSA and covered with a thin layer of ProLong Gold. Cover slips were then applied before being stored in the dark at −20 °C until sections were examined. Representative staining of subcellular characteristics of active zones and opposing ACh receptors are found in Figure 3. 

### 2.3. Immunofluorescent Staining of Myofibers

In order to quantify myofiber profiles, 10 µm thick cross-sections were obtained from the midbelly of the muscle using a cryostat set at −20 °C. Sections were then rinsed in PBS with 1% BSA for 5 min. All primary antibodies used to determine myofiber type via myosin heavy chain isoform expression were purchased from the Developmental Studies Hybridoma Bank at the University of Iowa. These antibodies were donated to that facility by Dr. Stefano Schiaffino of the University of Padova. The BA-D5 immunogen diluted 1:10 in PBS with 1% BSA was used to identify Type I myofibers, while SC-71 used at a concentration of 1:1 reacted specifically with Type IIA myofbers, and the F3 antibody diluted 1:1 visualized Type IIB myofibers. Type IIX myofibers were identified by their lack of immunofluorescence, although they were still visible. After adding the described diluted primary antibodies, muscle sections were incubated in humidified chambers at 37 °C for 1 h. Subsequently, sections were rinsed 3 × 5 min in PBS with 1% BSA and then incubated with fluorescently labeled secondary antibodies diluted 1:500 in PBS containing 5% goat serum. Secondary antibodies were conjugated to Alexafluor 555 (red fluorochrome) to identify Type I myofibers, AlexaFluor 350 (blue fluorochrome) to detect Type IIA myofibers, and AlexaFluor 488 (green fluorochrome) to visualize Type IIB myofibers. Following a 30 min incubation at 37 °C in a humidified chamber, sections were rinsed 3 × 5 min in PBS with 1% BSA before receiving a 3 min rinse in deionized water. Excess water was then blotted off the slide, and ProLong Gold was applied to muscle sections before applying cover slips and storing the slides in the dark at −20 °C until they were used for analysis. Immunofluorescent staining of myofibers in the EDL, plantaris, and soleus can be viewed in Figure 4.

### 2.4. Microscopy

An Olympus FV 300 confocal system featuring three lasers and equipped with an Olympus BX60 fluorescent microscope (Olympus America, Center Valley, PA, USA) was used to collect and store images of NMJs. Using a 100× oil immersion objective, it was initially established that the entire NMJ was within the longitudinal borders of the myofiber. This ensured an “en face” orientation before taking measurements and that no damage to the myofiber structure had occurred during tissue sectioning. Then a detailed image of the entire NMJ structure was constructed from a z-series of scans taken at 0.5 µm thick increments. Digitized, two-dimensional images of entire NMJs were stored on the system’s computer hard drive and were later quantified with Image-Pro Plus software (Media Cybernetics, Rockville, MD, USA). In each muscle, 10–12 NMJs were quantified, and measurements were averaged to represent NMJ morphology within that muscle. 

In the assessment of myofiber profiles, an Olympus BX41 microscope (Olympus America, Center Valley, PA, USA) equipped with fluorescence capacity (X-Cite, Excelitas Technologies, Waltham, MA, USA) was used in conjunction with Infinity Analyze software (Lumenera Corporation, Ottawa, Canada). A random sample of 125–150 myofibers from each muscle was analyzed to determine the average myofiber size (cross-sectional area) and fiber type composition within that muscle. 

### 2.5. Statistical Analysis

All results are reported as means ± SE. Each individual variable assessed for differences between soleus, plantaris, and EDL muscles were conducted so with one-way ANOVA. In the event of a significant (*p* ≤ 0.05) F ratio, a Tukey post hoc analysis was conducted to evaluate pairwise comparisons. A sample size of 10–12 NMJs/muscle was selected as numerous other studies from our lab have shown this to be adequate for the desired power of 0.8. In all cases, the normality of data distribution conformed with the Shapiro-Wilk test.

## 3. Results

The average body mass for the 10 adult male Wistar rats used as subjects for this study was 358.7 ± 3.4 g which is normal for this strain of rat [12].

### Cellular NMJ Results

In examining results from data collection on gross or cellular aspects of NMJ morphology, it was found that there was a significant between muscle difference in the number of pre-synaptic nerve terminal branches. Specifically, the number of branches was higher in nerve terminals residing in the soleus than it was in nerve terminals in both the plantaris (*p* < 0.005), and the EDL (*p* = 0.003), with no significant difference between the EDL and plantaris. Yet when looking at the total length of nerve terminal branching, it did not closely mimic differences in branch number. That is, while it was discovered that the soleus did have a greater total branch length than both the plantaris and the EDL, post-hoc analysis showed that it reached the level of statistical significance only when the soleus was compared to the plantaris (*p* < 0.001); when compared to the EDL, the branch length of the soleus only approached significance (*p* = 0.069). A significant difference in total branch length did exist between the EDL and the plantaris, i.e., EDL > plantaris, however (*p* = 0.002). And when examining average branch length, i.e., total branch length/branch number, it was determined that a significant (*p* = 0.028) ANOVA generated F ratio between muscle difference was apparent. Post-hoc analysis revealed that the average length of EDL branches significantly exceeded both that of the soleus (*p* = 0.027) and the plantaris (*p* = 0.014), with no difference between the soleus and plantaris. When comparing branching complexity between those three muscles, a significant (*p* = 0.008) ANOVA result was noted, and post-hoc analysis indicated that, similar to branch number, soleus values exceeded both those of the plantaris (*p* = 0.003) and the EDL (*p* = 0.025), and that no significant difference was identified between the plantaris and the EDL. 

Results gathered from quantifying pre-synaptic vesicles and pre-synaptic nerve terminal branch length was then used to provide insight into the ability of nerve terminal branches to express ACh-containing vesicles. Initial ANOVA results underscored a muscle difference in this important variable (*p* = 0.050), thus enabling a post-hoc analysis. This showed a single significant (*p* = 0.018) difference, and it was that the “packing” of vesicles on a given length of nerve terminal branch was higher (~44%) in the soleus than it was in the EDL. This variable was similar in the soleus and the plantaris (*p* = 0.525) and the plantaris and the EDL (*p* = 0.078).

In examining exclusively immunofluorescent staining of pre-synaptic vesicles, the manually drawn perimeter encircling the entire expression of stained vesicles along with unstained interspersing regions (total perimeter) revealed no between muscle difference (*p* = 0.166). However, when cumulative perimeter length encasing only stained clusters of vesicles but not the unstained areas between clusters (stained perimeter) was assessed, ANOVA results displayed a significant (*p* = 0.001) between muscle differences. When post-hoc procedures were conducted, it was determined that the length of the vesicle staining perimeter was higher in the soleus than in both the plantaris (*p* = 0.001) and the EDL (*p* = 0.007). 

Another informative measure of pre-synaptic vesicle expression is the area, rather than the perimeter length, of vesicle staining as it approximates the number of vesicles present. When viewing the area of staining as the total area within the perimeter length manually drawn around the entire area of vesicles, i.e., including unstained areas between stained receptor clusters, initial ANOVA analysis yielded a statistically significant (*p* = 0.020) difference. The subsequent post-hoc Tukey test presented a single significant pairwise difference (*p* = 0.006) where the soleus’ manually drawn area was greater than that of the plantaris. And when examining solely the automatically calculated area stained for vesicles, not including unstained regions between stained vesicle clusters, i.e., stained area, a significant F ratio (*p* = 0.000) from ANOVA procedures was noted. The post-hoc procedure to examine pairwise comparisons showed that the area of pre-synaptic vesicular staining was significantly greater in the soleus than it was both in the plantaris (*p* = 0.000) and the EDL (*p* = 0.000). However, vesicular stained areas were almost identical in the EDL and plantaris (*p* = 0.858). Finally, the dispersion of pre-synaptic vesicles was assessed, but initial ANOVA results failed to identify significant results for that variable (*p* = 0.335). 

Since cellular or gross aspects of NMJ structure also include the post-synaptic endplate region harboring ACh receptors, characteristics of BTX reactivity were also examined. The first variable assessed was manually drawn total perimeter length encompassing ACh receptor clusters and unstained regions interspersed between those clusters. ANOVA results of the length of this manually drawn line revealed a significant (*p* = 0.000) F ratio enabling a post-hoc pairwise analysis. This, in turn, produced a significant (*p* = 0.000) and 3-fold greater value in the soleus relative to both the plantaris and the EDL, with no difference between the plantaris and the EDL. Similar to vesicle immunofluorescent staining, the Image-Pro Plus software was used to automatically draw lines encompassing the stained receptor clusters—without including interspersed unstrained areas—and again, a significant (*p* = 0.008) F-ratio from the initial ANOVA procedure was noted for this stained perimeter length. The subsequent post-hoc analysis showed that, as with manually drawn perimeters, the automatically drawn perimeter lengths capturing solely the stained receptor clusters were significantly larger in the soleus than in the plantaris (*p* = 0.012) and the EDL (*p* = 0.004).

When quantifying the area of post-synaptic endplate regions, again manually drawn lines encompassing the entire endplate region and lines drawn automatically and exclusively around stained receptor clusters were used to calculate the area. In examining the total area enclosed by manually drawn lines, initial ANOVA results did not identify a significant F-ratio (*p* = 0.494), precluding analysis of pairwise differences. However, when stained areas within automatically drawn lines were assessed, a significant (*p* = 0.003) F-ratio was detected. The ensuing post-hoc analysis revealed that the area exclusively stained with BTX on endplates of the soleus was significantly greater than the stained endplate area of the plantaris (*p* = 0.001) and the EDL (*p* = 0.023), with no significant difference between plantaris and EDL endplates. Similar to pre-synaptic vesicles, the dispersion of ACh receptors at the post-synaptic endplate was also quantified. Initial ANOVA results displayed a significant (*p* = 0.000) F-ratio, thus enabling pairwise comparisons. Those comparisons indicated that the clusters of ACh receptors were significantly more compact and less dispersed in the soleus than in both the EDL (*p* = 0.002) and the plantaris (*p* = 0.000), with no difference between the plantaris and EDL (*p* = 0.210).

Overall then, when examining cellular features of NMJs in EDL, soleus, and plantaris muscles, it was noted that overwhelmingly, both pre- and post-synaptic features of the NMJ were significantly larger in the soleus compared to EDL and plantaris muscles. Indeed, of the 15 NMJ characteristics assessed, 10 of them were significantly larger in the soleus than in the plantaris or EDL. This is despite the fact that the average myofiber size (cross-sectional area) was smaller in the soleus than in the EDL and the plantaris. All results concerning cellular NMJ data analyses can be found in Table 1.

Of equal importance to this project was the analysis of subcellular components of the NMJs of the three muscles of interest. More specifically, examination of the critical pre-synaptic active zone was considered to be a priority. The first variable assessed was the stained dimensions of Bassoon, an essential scaffolding protein of the active zone. Initial ANOVA results noted a significant (*p* = 0.000) F-ratio for the manually drawn length of Bassoon staining. Subsequent post-hoc results identified that active zone length was much (~3-fold) more impressive in the soleus than in both the plantaris (*p* = 0.000) and the EDL (*p* = 0.000). Surprisingly, when the length of Bassoon staining was determined with an automatically generated line via software, the initial F-ratio was again significant (*p* = 0.000), but it was the plantaris that was found to have larger active zones than both the EDL (*p* = 0.000) and the soleus (*p* = 0.000) with no difference between those two muscles (*p* = 0.586). Regarding the area of Bassoon staining, assessment for the total area of active zones—both stained and unstained interspersing regions—the initial ANOVA displayed a strong similarity (*p* = 0.993) between muscles. However, when examining the cumulative area of stained regions of Bassoon—discounting unstained interspersed regions—assessed via software, initial ANOVA results produced a significant (*p* = 0.043) F-ratio permitting follow-up pairwise comparisons. Post-hoc results showed that plantaris active zones were significantly smaller than those of the soleus (*p* = 0.021) and the EDL (*p* = 0.042), with no difference between the active zones of the soleus and the EDL (*p* = 0.787). Dispersion of Bassoon staining was then examined with the initial ANOVA procedure generating a significant (*p* = 0.001) effect allowing post-hoc analysis. This, in turn, showed that the dispersion of Bassoon staining was greater, i.e., less compact, in the plantaris than it was in the soleus (*p* = 0.000) and the EDL (*p* = 0.004).

The next subcellular variable assessed was the P/Q variant of the calcium channels that must reside in close proximity to the docked vesicles located at the active zone in order to stimulate exocytosis of those vesicles. Initially, the manually drawn length of P/Q staining (stained clusters plus unstained interspersed regions between clusters) was analyzed with a significant (*p* = 0.000) F-ratio being generated with ANOVA. The ensuing post-hoc analysis showed that the P/Q calcium channel staining length in the soleus was more than twice that for either the EDL (*p* = 0.000) or the plantaris (*p* = 0.000). The F-ratio derived from the ANOVA comparing the length assigned by the software package that captured only stained zones was again noted as being significant (*p* = 0.022). Post-hoc results showed the EDL to be significantly less in length than the soleus (*p* = 0.010) and the plantaris (*p* = 0.027); no difference was discerned between the soleus and the plantaris (*p* = 0.697). 

Next to be evaluated was the total area of P/Q channels (stained clusters along with unstained interspersed regions), and initial ANOVA results failed to note a significant F-ratio (*p* = 0.774). In contrast, stained areas expressing calcium channel staining and excluding interspersed unstained areas showed a significant (*p* = 0.008) F-ratio, allowing analysis of pairwise differences. These post-hoc procedures indicated that the expression of P/Q channels was significantly greater in the soleus than it was in both the EDL (*p* = 0.013) and the plantaris (*p* = 0.003). Also quantified was the dispersion of P/Q calcium channels expressed at active zones. The ANOVA findings pointed to a significant (*p* = 0.000) F-ratio leading to pairwise comparisons via post-hoc procedures. These, in turn, indicated that the expression of calcium channels was significantly more compact in soleus muscles than in EDL (*p* = 0.001) and plantaris muscles (*p* = 0.000). 

Essential to the proper functioning of the NMJ is the proper alignment of pre-synaptic vesicle release sites with post-synaptic binding sites. To assess this, the coupling of pre-synaptic Bassoon with post-synaptic BTX was quantified. Initial ANOVA results detected a significant (*p* = 0.001) F-ratio allowing post-hoc assessment where it was shown that Bassoon to BTX coupling was more tightly matched, i.e., closer to 1:1 ratio, in the EDL than in the soleus (*p* = 0.001) or plantaris (*p* = 0.000), with no difference between the soleus and plantaris (*p* = 0.886). The coupling of P/Q calcium channels expressed at active zones with post-synaptic BTX staining was also similarly quantified. Initial ANOVA procedures did not identify a significant (*p* = 0.331) F-ratio, indicating an absence of between-muscle differences. Finally, the physical relationship between P/Q channels and the Bassoon cytoskeleton of the active zone was examined. Initial ANOVA results showed a significant (*p* = 0.007) F-ratio necessitating subsequent post-hoc procedures, which noted that the ratio of calcium channel expression relative to active zone area was significantly less in the EDL than it was in the soleus (*p* = 0.004) and the plantaris (*p* = 0.007), with no difference, observed between the soleus and the plantaris (*p* = 0.918). 

When examining subcellular aspects of NMJ morphology—mainly seen at pre-synaptic active zones—data again showed that areas containing vesicular release sites were larger in the soleus than in the plantaris or EDL muscles. However, muscle differences were less pronounced in subcellular than cellular morphology. That is, only four of the 13 (31%) subcellular variables assessed displayed larger sizes in the soleus compared to 10 of the 15 cellular features (67%). All data regarding subcellular evaluations are presented in Table 2.

Also deemed important to the present project were myofiber profile analyses of the same muscles used to determine cellular and subcellular characteristics of NMJs. The first myofiber variable to be examined was average myofiber size, i.e., cross-sectional area, without distinguishing among fiber types. Results of the initial ANOVA provided a significant (*p* = 0.000) effect, which was then followed up with the Tukey post-hoc to identify specific pairwise differences. Results from post-hoc tests highlighted the presence of three separate pairwise differences whereby each muscle was different from both of the other muscles evaluated. That is, plantaris myofibers were significantly (*p* = 0.002) larger than soleus muscles, which were significantly (*p* = 0.012) larger than EDL fibers. 

Next, the myofiber size for each fiber type was analyzed individually, beginning with Type I fibers. Results on the cross-sectional area for these slow-twitch fibers started with a significant (*p* = 0.000) F-ratio for the initial ANOVA procedure. As a result, post-hoc procedures were conducted to determine pairwise differences between the three muscles of interest. The outcome was the same as when fiber types collapsed together in that all three muscles were different from each other (*p* = 0.000). However, unlike data gathered when collapsing fiber types together, when examining Type I fibers alone, it was ascertained that those slow-twitch fibers of the soleus were larger than those of the plantaris (*p* = 0.000), which, in turn, were larger than those expressed in the EDL (*p* = 0.001). And when examining cross-sectional areas solely of Type IIA fibers, the initial ANOVA yielded a significant (*p* = 0.001) F-ratio with post hoc results showing that those fibers were smaller in the EDL than in both the soleus (*p* = 0.000) and the plantaris (*p* = 0.000) but with no difference between soleus and plantaris Type IIA fibers (*p* = 0.506). Next, Type IIX myofibers were evaluated, and an initial ANOVA procedure produced a significant (*p* = 0.001) F-ratio. In subsequent post-hoc procedures, pairwise differences showed only a single significant difference in that plantaris Type IIX myofibers were significantly (*p* = 0.000) larger than those same fibers within the EDL. Finally, when assessing only Type IIB myofibers, the initial ANOVA test produced a significant (*p* = 0.000) F-ratio. Post-hoc analysis revealed that these fast-twitch, glycolytic fibers [8,13,14] were larger in the plantaris than the EDL (*p* = 0.002). As previously reported [8], no type IIB fibers are expressed in the soleus.

The other facet of the myofiber profile of a muscle is myofiber-type composition. When comparing the Type I fiber composition of the three muscles of interest, a significant (*p* = 0.000) F-ratio for the ANOVA was first established. Subsequent post-hoc analysis indicated three significant pairwise differences whereby the percentage of Type I fibers in the soleus was significantly greater than that of the plantaris (*p* = 0.000) and the EDL (*p* = 0.000), and the slow-twitch fiber composition of the plantaris was greater than that of the EDL (*p* = 0.017). Again, the initial ANOVA output for comparing the contribution of Type IIA myofibers was significant (*p* = 0.028), enabling post-hoc analysis. Results of that Tukey analysis revealed that the percentage of Type IIA myofibers in the soleus was significantly less than in the plantaris (*p* = 0.028), as well as the EDL (*p* = 0.016), with no significant difference between EDL and plantaris muscles (*p* = 0.816). Next, the composition of Type IIX myofibers was determined. Again, the initial ANOVA procedure evoked a significant (*p* = 0.011) F-ratio, allowing pairwise comparisons. Those comparisons indicated that the percentage of Type IIX fibers in the soleus was significantly less than in the plantaris (*p* = 0.009) as well as in the EDL (*p* = 0.003), with no difference between the EDL and plantaris (*p* = 0.265). Finally, Type IIB myofiber type composition was quantified, producing a significant (*p* = 0.030) F-ratio in the ANOVA analysis, but without a significant difference between the plantaris and EDL (*p* = 0.105). Recall that Type IIB fibers are not expressed in the soleus of the rat, so not pairwise comparisons could be made with that muscle. Data concerning myofiber profiles are displayed in Table 3. 

## 4. Conclusions

The neuromuscular system is one of the largest in the human body, accounting for roughly 40% of total body mass [15]. In addition to its size, that system is invaluable in that it plays a role in the body’s overall health by contributing to blood sugar, thermoregulatory, endocrine, and even bone homeostasis [15]. Given its large size and an array of physiological responsibilities, it comes as no surprise that communication between the neural and muscular components of the neuromuscular system is tightly controlled and maintained even as activation of the neuromuscular system varies widely, i.e., sedentary, passive behavior through high-intensity exercise. At the very core of this regulation of neuromuscular function is the NMJ, as it enables the communication between the motor neuron and its myofibers. In order to gain a closer look at how the integrity of neuromuscular function might be maintained; this project examined a host of morphological traits of the neuromuscular system. This focus included pre- and post-synaptic elements of the NMJ, both at the cellular and subcellular levels, as well as the profiles of the myofibers innervated by NMJs. To better understand the factors that may account for optimal neuromuscular function, it was necessary to examine a range of muscles that differed by function, myofiber type composition, and recruitment patterns, i.e., duty cycle. The muscles that provided this spectrum of features were the soleus which is a slow-twitch ankle extensor, and the main postural and locomotor muscle with a high-duty cycle. The plantaris, a predominantly fast-twitch ankle extensor, locomotor, but not postural, muscle with a moderate duty cycle, and the EDL which is primarily a fast-twitch ankle flexor muscle, which significantly contributes to neither posture nor locomotion, thus having a light duty cycle. 

This broad scope of variables observed provided an equally wide array of data showing that, above all else, it appears that recruitment patterns are the factor that most acutely influences NMJ and myofiber structural characteristics. For example, when examining cellular features of the NMJ, such as pre-synaptic nerve terminal branching patterns and vesicle content, it was the highly recruited soleus that stood out among the three muscles of interest. Importantly, it was determined that the capacity to express vesicles per given unit of branch length was 44% greater in the soleus. This is significant in that the potential to store that much more ACh in the pre-synaptic soleus indicates a greater ability to sustain prolonged trains of stimuli and thus display impressive neuromuscular endurance when called upon to do so. It is reasonable to conclude that this, at least in part, explains why muscle contractions during a long train of stimulation to the soleus continue for longer durations and with modest declines in force production compared to other skeletal muscles [16,17,18,19]. In addition to being able to more effectively store ACh-laden vesicles per unit length in the pre-synaptic branches of the soleus, the total number of vesicles attached to the nerve terminal branches of the soleus was elevated compared to the other muscles examined. This is best explained not only by the greater number of vesicles per given length of terminal branching but also by the significantly greater total length of branching in the soleus relative to the other muscles. This greater total content of vesicles stored in the soleus gives it an advantage when being subjected to a prolonged train of stimuli which characterizes the demands of that postural and locomotor muscle [5,6,14]. 

The project’s post-synaptic investigation at the cellular level indicated that the total perimeter length capturing ACh receptors and the area of staining of those receptors was greater in the soleus than in the EDL and the plantaris. Moreover, staining for ACh receptors was established as being more compact, i.e., less dispersed, in the soleus than it was in the other two muscles examined here. This more compact arrangement suggests a more prolific ability to support long-lasting neuromuscular transmission and muscle activation. In effect, then, it was revealed that endplate regions of the soleus were morphologically distinct from those of the EDL and plantaris, while there were no consistent differences between the EDL and plantaris muscles. 

Perhaps the reason that the NMJs of the soleus is larger than those of the other muscles examined is related to the fact that Type I, or slow-twitch, myofibers are the most abundant in that muscle, in contrast to the plantaris and EDL, where Type II or fast-twitch myofibers are the dominant types. However, previous investigation [20] revealed that NMJs associated with Type I myofibers are smaller, not larger, than those identified on Type II myofibers. Thus, differences in myofiber-type composition are not likely to explain muscle-specific differences in NMJ dimensions observed in the current study. Rather, differences in recruitment patterns closely matched muscle-specific NMJ dimensions, and it is more reasonable that patterns of usage of a muscle rather than its fiber type composition, or function as a flexor or extensor, account for NMJ size differences among the muscles examined here.

Since it has been reported that subcellular components, or active zones, of the NMJ play a more important role in synaptic transmission than overall nerve terminal size [21], specific structural measures of active zones were also assessed in the soleus, plantaris, and EDL muscles, the two features of the active zones that were ascertained were the structural protein Bassoon which serves as the physical foundation of the active zones that dock ACh containing vesicles [22,23], and the P/Q variant of the calcium channels that reside in close proximity to the vesicles. This position is maintained so that calcium entering the cytomatrix of the terminal ending upon opening of the channels will release the vesicles from their docked positions and enable exocytosis of ACh stored by vesicles into the synaptic cleft preceding neuromuscular transmission [23,24]. Indeed, the proximity of those channels to vesicles is necessitated by the fact that the chances of a vesicle actually undergoing exocytosis in response to a single neural impulse of the nerve terminal is very low, even less than 10% [24,25,26]. Another important benefit of this positioning of the P/Q channels is that proper alignment of active zones, i.e., in apposition from the post-synaptic ACh receptors, is assured. This is made possible because the channels are bound to the laminin component of the endplate, extending across the synaptic cleft before fixing themselves nearby the docked vesicles at the active zone, thus stabilizing pre- to post-synaptic coupling [22,27].

In the current study, it was noted that the size of Bassoon-stained active zones had a greater manually drawn perimeter length in the soleus than in both the plantaris and the EDL. In contrast, the automatically derived cumulative perimeter length was almost twice as long in the plantaris than in both the soleus and EDL. However, when examining the size of the active zone as the cumulatively stained area of ACh receptor clusters, it was documented that the active zones of the plantaris were about one-half the size of those of the soleus and the EDL. In a likewise fashion, the dispersion of Bassoon staining was more pronounced, and thus less compact, among the EDL and soleus than it was in the plantaris. It is quite possible that the two locomotor muscles, i.e., soleus, and plantaris, require more compact active zones and vesicle docking zones than the less recruited EDL.

When examining the calcium channels located at the active zones, it was noted that the manually drawn line encompassing staining of P/Q channels and their interspersing regions, i.e., total length, was roughly twice as long in the soleus than it was in the EDL and plantaris. This pattern did not hold, however, when quantifying automatically drawn perimeter length around P/Q channels while excluding unstained interspersed regions, where the three muscles of interest showed similar results for stained perimeter length. Next to be considered was area of staining, and for this, it was determined that the automatically derived area of calcium channel staining was greater in the soleus than it was in both the plantaris and the EDL. And when quantifying the dispersion of P/Q channel staining, it was again revealed that, as with Bassoon staining, channel expression was more compact in soleus muscles than it was in both plantaris and EDL muscles. This suggests that a more condensed and powerful bolus of calcium enters the nerve terminal near the active zones upon arrival of the action potential descending down the motor neurons axon in the highly active soleus. This would, of course, increase the probability of ACh release from the vesicles. This would be meaningful given the slight chance (<10%) of any single vesicle being released from its docked site in response to a single electrical impulse delivered to the terminal’s bouton [24]. 

This project also assessed pre- to post-synaptic coupling by quantifying overlap between vesicular release sites as determined with Bassoon staining with post-synaptic receptors indicated by BTX staining. Here, it was uncovered that pre- to post-synaptic coupling was more impressive, i.e., closer to a 1:1 ratio, in the EDL than it was in the soleus and the plantaris muscles where BTX area exceeded the Bassoon area by roughly 45%. Since both the soleus and plantaris are integral locomotor muscles, while the EDL is not [6,28], it is reasonable to conclude that the greater and more severe recruitment received during ambulation necessitates an increased number of post-synaptic receptors relative to docked vesicles during prolonged, continuous stimulation. This is unsurprising since, during prolonged stimulation, those receptors would experience desensitization [29,30]. However, with excess receptors, effective neurotransmission continues, especially since pre-synaptic vesicles undergo recycling and can continue to participate. 

On a related note, it was determined that the coupling of calcium channels with Bassoon stained active zones was significant of greater accuracy in the EDL, i.e., a perfectly matched 1:1 ratio, than in the soleus and plantaris where once again Bassoon staining, or active zones, exceeded the area of calcium channels. It is possible that the excess receptor area relative to the active zone area is linked to the greater recruitment patterns of the soleus and plantaris—both are principal ambulatory muscles—relative to the EDL with its low recruitment patterns and need for only periodic calcium influxes. 

The NMJ is designed to elicit skeletal muscle contraction, and the muscle component of the neuromuscular system certainly accounts for the greatest portion of that system. Indeed, the endplate of the NMJ occupies less than 1% of the sarcolemma of any muscle fiber [25,26,31]. Given this, it was also deemed important that skeletal muscles innervated by the NMJ be quantified via myofiber profile assessment. In large part, these analyses produced predictable results. For example, myofibers of the plantaris—with no differentiation of fiber type—were the largest of the muscles assessed, while those of the soleus were larger than those of the EDL, as has been previously reported [8,13]. Moreover, the size of specific myofiber types was found to be similar to earlier research in that in the soleus, Type I fibers displayed the largest cross-sectional areas, while Type IIA fibers of the soleus and plantaris were both larger than those of the EDL [8]. Type IIX myofibers were once again the largest in the plantaris, as were Type IIB fibers. It was interesting to note the disconnect between NMJ size and myofiber size in these results. For example, the plantaris displayed the largest myofiber size, but the soleus exhibited the largest endplate area. Previous studies have reported that endplate size was dependent on the size of associated myofibers [32,33], but that was applicable to changes associated with natural growth and development. In fully mature and developed animals, no such relationship exists. The myofiber composition of the soleus, plantaris, and EDL muscles examined here was consistent with what has been reported earlier [8,13]. That is, Type I fibers are the most predominant in the soleus, while Type II fibers dominate in both the plantaris and EDL.

When revisiting this project’s initial query about what may affect NMJ size in various muscles displaying differing characteristics, it appears that recruitment patterns of specific muscles influence cellular and subcellular NMJ morphology most powerfully. In several instances, including the ratio of BTX to Bassoon area, P/Q channel to Bassoon ratio, and P/Q channel perimeter, NMJ structure was similar between the soleus and plantaris muscles but different from features noted in the EDL. Recall that the plantaris and soleus muscles are highly recruited as locomotor muscles while the EDL is not [6,28]. Further support for the contention that duty cycle matters most significantly in affecting NMJ morphology is evidence that in several other structural parameters such as BTX area, terminal branch number, and pre-synaptic vesicle content, data values collected from the soleus exceeded those taken from both the EDL and the plantaris.

The present project also found that unlike an earlier study [20], the data reported here suggest that the size of underlying myofibers does not determine the size of NMJs. Here, the smallest myofibers (soleus) exhibited the largest NMJs. It should be noted, however, that the report indicating that myofiber size influenced NMJ size examined the three major myofiber types on the same muscle (diaphragm), which is heterogenous regarding myofiber type expression. This differs from the present study with its use of three disparate muscles (soleus, EDL, plantaris). Indeed, the data presented here do not support the suggestion that the size of the NMJ is affected by underlying myofiber type since fiber type distribution in the EDL and plantaris muscles examined were quite similar, while NMJ structure was not. This is likely related to the fact that those two muscles, i.e., the EDL and plantaris, have unique duty cycles and recruitment patterns despite similar myofiber-type profiles. That result is not too surprising since it has been shown that different recruitment patterns imposed by exercise training and disuse alter NMJ structure [3,4]. 

All told, the results of this investigation of muscle specificity of neuromuscular system morphology demonstrate that cellular and subcellular NMJ characteristics differ among muscles with distinct recruitment profiles. That said, generally, it is true that cellular and subcellular features of the NMJ mirror each other. Moreover, different components of active zones display unique or muscle-specific features and pre- and post-synaptic components of the NMJ are typically, but not always, well-coupled. Finally, muscle-specific distinctions in myofiber profile may not always be emulated in NMJ structural features. These observations serve to underscore the keen sensitivity of the neuromuscular system to specific demands imposed upon it.

## Figures and Tables

**Figure 1 cells-12-00361-f001:**
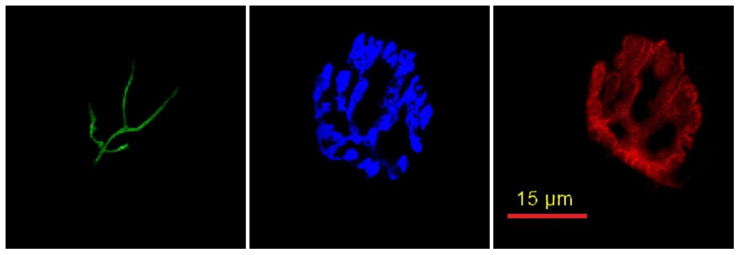
Representative immunofluorescent staining cellular features of the NMJ. Pre-synaptic nerve terminal branches are highlighted with AlexaFluor 488 (green), pre-synaptic acetylcholine-containing vesicles are displayed with AlexaFluor 647 (blue), and post-synaptic acetylcholine receptors are visualized with rhodamine (red).

**Figure 2 cells-12-00361-f002:**
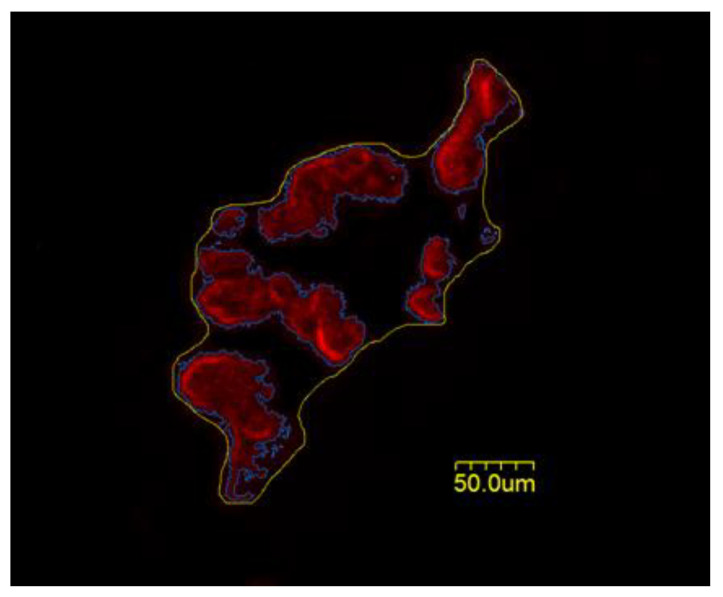
Representative image of tracings used to quantify post-synaptic endplate morphology. The outermost line encompassing the endplate (yellow) is manually drawn, including both stained receptors and empty regions between clusters, and used to quantify “Total Perimeter” and “Total Area,” while the innermost line (blue) is automatically drawn around clusters of ACh receptors excluding interspersed empty areas, with Image-Pro Plus software and used to quantify “Stained Perimeter” and “Stained Area.” Scale bar = 50 um.

**Figure 3 cells-12-00361-f003:**
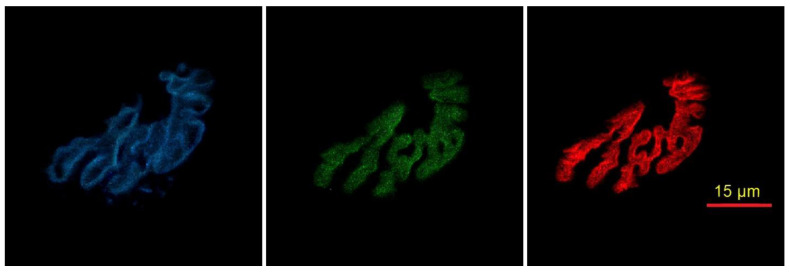
Representative immunofluorescent staining of subcellular features of the NMJ. Pre-synaptic active zone P/Q calcium channels are identified with AlexaFluor 647 (blue), pre-synaptic Bassoon is identified with AlexaFluor 488 (green), while associated post-synaptic acetylcholine receptors are visualized with rhodamine (red).

**Figure 4 cells-12-00361-f004:**
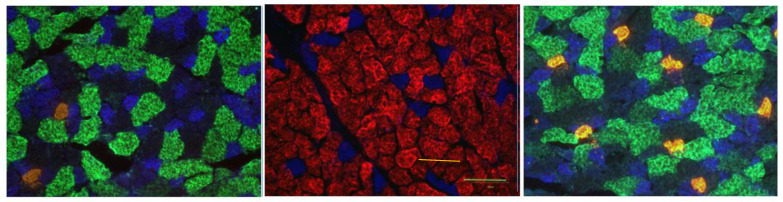
Representative immunofluorescent staining of myofibers. The first panel is an EDL muscle, the middle panel is of the soleus muscle, and the third panel is of plantaris muscle. In each muscle portrayed, Type I fibers are identified with AlexaFluor 555 (red), Type IIA fibers are stained with AlexaFluor 350 (blue), Type IIB fibers are identified with AlexaFluor 488 (green), while unstained fibers (with cell borders still visible) are those of the Type IIX category. Scale bar in center panel = 100 µm.

**Table 1 cells-12-00361-t001:** Cellular morphology of NMJs in muscles of interest.

Variable	EDL	Soleus	Plantaris
Nerve terminal branch number	5.9 ± 0.3	7.6 ± 0.3 *	6.0 ± 0.6
Total branch length (µm)	129.8 ± 1.0	145.6 ± 0.6	100.4 ± 5.9 *
Average branch length (µm^2^)	22.9 ± 29.4 *	19.5 ± 0.6	18.9 ± 1.4
Branching complexity	8.5 ± 0.8	11.7 ± 0.8 *	7.1 ± 1.3
Vesicle/Branch length	0.667 ± 0.08	0.959 ± 0.07 ^†^	0.884 ± 0.09
Total length around vesicles (µm)	140.8 ± 5.2	163.5 ± 10.8	143.7 ± 9.5
Stained vesicle perimeter (µm)	180.1 ± 14.1	239.5 ± 18.4 *	160.3 ± 8.0
Total vesicle area (µm^2^)	421.9 ± 52.1	510.9 ± 33.7 ^‡^	351.5 ± 21.5
Stained vesicle area (µm^2^)	79.7 ± 9.1	138.3 ± 12.5 *	77.1 ± 6.0
Vesicle dispersion (%)	23.0 ± 2.6	27.8 ± 1.5	24.8 ± 2.7
Total BTX perimeter (µm)	95.3 ± 6.8	319.0 ± 17.7 *	90.6 ± 5.2
Stained BTX perimeter (µm)	224.5 ± 16.1	315.9 ± 19.9 *	237.5 ± 25.3
Total BTX area (µm^2^)	372.5 ± 29.4	401.7 ± 24.8	354.8 ± 30.4
Stained BTX area (µm^2^)	191.2 ± 35.1	272.8 ± 17.1 *	145.7 ± 15.9
BTX dispersion (%)	49.1 ± 5.9	67.8 ± 1.6 *	42.1 ± 2.8

Values are means ± SE; N = 10/muscle; Abbreviations: BTX = post-synaptic α-bungarotoxin; Branching complexity = branch number × total branch length ÷ 100; Dispersion = stained area ÷ total area × 100; higher values equal greater compactness of staining; * indicates significant (*p* ≤ 0.05) difference from both other muscles; ^†^ indicates significant (*p* ≤ 0.05) difference from EDL; ^‡^ indicates significant (*p* ≤ 0.05) difference from plantaris.

**Table 2 cells-12-00361-t002:** Subcellular morphology of NMJs in muscles of interest.

Variable	EDL	Soleus	Plantaris
Total Bassoon perimeter (µm)	91.3 ± 7.5	282.8 ± 16.9 *	86.9 ± 4.2
Stained Bassoon perimeter (µm)	328.6 ± 49.9	359.8 ± 23.4	605.1 ± 45.8 *
Total Bassoon area (µm^2^)	362.4 ± 38.8	357.9 ± 24.4	358.4 ± 29.6
Stained Bassoon area (µm^2^)	180.4 ± 39.6	189.8 ± 13.8	104.9 ± 11.4 *
Bassoon dispersion (%)	46.6 ± 6.3	53.3 ± 2.3	27.9 ± 1.5 *
Total P/Q perimeter (µm)	91.4 ± 4.9	221.8 ± 17.7 *	90.6 ± 5.2
Stained P/Q perimeter (µm)	230.0 ± 11.1 *	319.0 ± 22.4	306.4 ± 30.3
Total P/Q area (µm^2^)	377.5 ± 30.7	369.5 ± 47.8	409.9 ± 43.4
Stained P/Q area (µm^2^)	172.9 ± 35.2	274.9 ± 27.2 *	151.8 ± 16.3
P/Q dispersion (%)	42.6 ± 5.5	80.6 ± 7.8 *	38.6 ± 2.1
BTX stained area/Bassoon stained area	0.92 ± 0.10 *	1.45 ± 0.05	1.43 ± 0.13
P/Q stained area/Bassoon stained area	1.00 ± 0.07 *	1.48 ± 0.13	1.46 ± 0.10
BTX stained area/P/Q stained area	1.25 ± 0.21	1.03 ± 0.06	0.97 ± 0.05

Values are means ± SE; N = 10/muscle. Dispersion = stained area ÷ total area × 100; higher values equal greater compactness of staining. * indicates significant (*p* ≤ 0.05) difference from both other muscles.

**Table 3 cells-12-00361-t003:** Myofiber profiles.

Variable	EDL	Soleus	Plantaris
*Cross-sectional area* (µm^2^)			
Types combined *	2181 ± 155	2862 ± 112	3720 ± 223
Type I *	1284 ± 61	3001 ± 121	2006 ± 170
Type IIA	1238 ± 56 ^‡^	2246 ± 94	2141 ± 155
Type IIX	2030 ± 125	2340 ± §	3776 ± 331 ^‡#^
Type IIB *	3063 ± 244	N/A	5289 ± 683
*Fiber type composition* (%)			
Type I *	6.0 ± 0.8	80.7 ± 1.3	10.2 ± 1.1
Type IIA	27.8 ± 3.0	18.1 ± 1.2 ^‡^	27.7 ± 2.8
Type IIX	30.5 ± 2.1	1.3 ± § ^‡^	29.3 ± 2.9
Type IIB	35.6 ± 4.5	N/A	32.7 ± 3.3

Values are means ± SE; N = 10/muscle. * indicates all pairwise differences were significant (*p* ≤ 0.05). ^‡^ indicates a significant (*p* ≤ 0.05) difference from other muscles. ^#^ indicates trend (*p* = 0.08) for the difference from soleus. § indicates a lack of SE due to the small number of that fiber type in the muscle. N/A indicates that the soleus in the rat does not express Type IIB fibers.

## Data Availability

Data made available upon reasonable request to corresponding author.

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
