# Peer review of "Cellular and Subcellular Characteristics of Neuromuscular Junctions in Muscles with Disparate Duty Cycles and Myofiber Profiles"

_cells, 2023, doi:10.3390/cells12030361_

Round 1

Reviewer 1 Report

The work of Dechenes and collaborators gives interesting analyses regarding major variances among muscles having different fiber type composition and duty cycle. Authors provide all the significant aspects related to the neuromuscular junction characteristics due to the specificity of the three analyzed muscles, using pertinent statistical analysis. The manuscript is fluid in term of English language and style, however a configuration in paragraph, each having a conclusion, would be even easier and agreeable to read.     

Author Response

The authors wish to thank the reviewer for investing the time and effort necessary to provide a thorough assessment of our manuscript. In our revised paper, we have made every effort to adequately address each of the concerns raised by the reviewer. We believe that in doing so, we have significantly improved the quality of our script and that it is now suitable for publication in Cells; we sincerely hope that the reviewer agrees.

  • Many thanks for the kind words, we agree with the reviewer’s suggestion to (if we correctly understand) conclude results sections for both cellular and subcellular NMJ features with a summery paragraph highlighting important results, and we have done so in the revised version of our paper.

Reviewer 2 Report

I was asked to review the manuscript of Deschenes et al. with the title "Cellular and Subcellular Characteristics of Neuromuscular Junctions in Muscles with Disparate Duty Cycles and Myofiber Profiles"

In principle, their study confirms expectations, namely that NMJ size correlates with activity pattern. Interestingly, it does not appear to correlate with muscle fiber diameter.

The writing style in some sentences is not very scientitic, p.ex. "climbing stairs",  
"carrying groceries", "dancing".... its suffcient to collectively call that 8physical activity]. The writing should be adjusted through the whole manuscript.

MAJOR

Altogether, I do not really catch the exciting news presented by this manuscript. It is a interesting piece of work, but should be published in a more specialized anatomical journal.

MINOR
- some skeletal muscles are abbreviated (EDL) others not? Please standardize and introduce abbreviations before use.
- Cryocut 1800, Reichrt-Jung, Nubloch, Germany....wrong spelling
- Figures are cited in Experimental Procedures. The authors appear not to be aware that Figures have to be cited in the Result part of the manuscript.
- Image analysis was not very understandable. On the other hand, these days algorithm like NMJ Analyzer , NMJ Morph etc are available and do not need to be reinvented.

Author Response

The authors wish to thank the reviewer for investing the time and effort necessary to provide a thorough assessment of our manuscript. In our revised paper, we have made every effort to adequately address each of the concerns raised by the reviewer. We believe that in doing so, we have significantly improved the quality of our script and that it is now suitable for publication in Cells; we sincerely hope that the reviewer agrees.

Major

The writing style has been amended so that it is more scientific in tone throughout the manuscript. Terms such as “climbing stairs” “carrying groceries” etc. have been eliminated.

Minor

  • Of the 3 muscles examined here, only the extensor digitorum longus (EDL) is routinely abbreviated (not the soleus or plantaris). As suggested, we have carefully spelled out each muscle’s name entirely and abbreviated it as the EDL at its first use (in abstract).
  • Thanks for catching tis, proper spelling of “Reichert Jung” has now been included.
  • Again, thank you for spotting this. In our revised script, all figures are cited in the Results section.
  • Apologies for not being clearer in our description of image analysis. Rather than NMJmorph, we use and excellent automated software package (Image Pro Plus) to perform measurements on standard morphological variables such as area, perimeter length, receptor/vesicle dispersion, we hope the reviewer will find this acceptable.

Reviewer 3 Report

The present manuscript addresses the correlation between muscle type, fiber type, NMJ morphological features, and pre- and postsynaptic components of the neuromuscular apparatus. Therefore, adult male rats were sacrificed, soleus, EDL, and plantaris muscles prepared, sliced in different modes, stained against markers of active zones, postsynaptic AChR, muscle fiber types. Microscopic image analysis was used assess correlations as described above. This study, albeit purely descriptive in nature, has the potential of serving as a relevant resource for NMJ research and it fills a gap of knowledge. However, some technical aspects and discussion points need to be addressed.

1. The manuscript relates the NMJ and fiber features to "recruitment" of muscles at several places. However, the introduction is very rudimentary and short and would massively profit from an overview of what is really known about differential muscle recruitment of muscles and of soleus, EDL, and plantaris in rats and other species.

2. This study largely relies on morphological features of NMJ, which were extracted by manual image analysis. However, in the past years, and generalized toolset, freeware, called NMJmorph has become a sort of reference for such analyses. To improve the comparability of research data and to render this manuscript the resource it could be, it would be highly recommended to use NMJmorph, here.

3. The number of NMJs and muscle fibers analyzed is too low. In particular, it is known that both show gradients of type and size from the rim to center in different muscles. Thus, it is imperative that such profiles are considered. This could be done, e.g., by analyzing always a quarter muscle slice. Likely, about 50 NMJs / muscle and a few hundred fibers per muscle would be more useful than the current data set.

4. The results part shows three tables, only. It would be very important (again in the sense of a future reference function) to show typical morphologies of fiber distributions, NMJs, and their subcellular features for each muscle and table.

5. In the discussion on lines 494ff. the logic is unclear to this referee. The authors state: "

Thus, differences in myofiber type composition are not likely to explain muscle specific differences in NMJ dimensions observed in the current study. Rather, differences in recruitment patterns closely matched muscle specific NMJ dimensions, and it is more reasonable that patterns of usage of a muscle rather than its fiber type composition, or function as a flexor or extensor, that account for NMJ size differences among the muscles examined here.

". For specific recruitment patterns (e.g., tonic or phasic), different fiber types are considered particularly useful (e.g., slow-oxidative vs. fast-glycolytic). Thus, the discussion as presented here is not easy to follow. Maybe, it could be helpful to link these assumptions to concrete correlations?

Minor:

1. Abstract: sentence on l. 25f. is incorrect.

2. Discussion: sentence l. 595-7 is unclear.

Author Response

The authors would like to first express their gratitude to the reviewer for taking the time and investing the effort in providing a thorough assessment of our manuscript, along with the kind introductory comments. Important and valid points in the further comments have also been made, in our revised script, we have made every effort to amply address each of those comments and in doing so have improved the quality of our paper. We sincerely hope that the reviewer agrees and will now find it suitable for publication in Cells.

  • This is a good point, we have delved back into the literature on muscle recruitment patterns to expand our background discussion (in Introduction) on duty cycles of muscles we used in the current study.
  • Another valid point is raised here, but rather than NMJ morph, we used an excellent imaging software program (Image Pro Plus) to automatically assess morphological features of the NMJ including endplate area, length of circumference around the endplate, density of staining for pre-synaptic vesicles, pre- to post-synaptic coupling, among others. The accuracy of this imaging system is comparable to NMJmorph. In addition to using Image Pro Plus to automatically determine these measurements, we also manually made those measurements using the computer’s muse. We agree with the belief that nuanced staining sometimes is best quantified with subjective, manual quantification. For sure, the most complete picture is derived by using both objective measurements performed by imaging software, along with subjective measurements gathered by manual use of a computer’s mouse.
  • We appreciate the point made here, but our laboratory has been using sample sizes of 10-12 NMJs to accurately represent all NMJs on that muscle for more than 20 years, and only after experimentally determining that 10-12 NMJs accurately represents all NMJs. Of course this is true only on a muscles as long as the 10-12 selected are free of damage and are examined in “en face” positioning. This was found even when as many as 50 per muscle were selected, in this case it was found that more was not necessarily better (at least not more than 10-12).
  • In our revised script, figures have been changed to show cellular (Fig 1) and subcellular (Fig 2) NMJ features in each of the 3 muscles studied, we agree that this is a better presentation of our findings.
  • We have re-phrased the wording on this topic to hopefully make the point more clearly to the reader. In essence, in trying to understand why NMJs of the soleus are larger than those of the plantaris or EDL, one thing that immediately jumps out is the high percentage of type I fibers in the soleus compared to the plantaris and EDL.But as reported in ref #20 (Prakash and Sieck) when examining a muscle with mixed fiber types i.e. the diaphragm, the NMJs of the soleus were found to be smaller, not larger, than type II fibers. Hence, the great preponderance of type I fibers in the soleus would not be expected to account for the larger synapses found in the soleus. Hopefully, this point has more clearly been made in our revised script.
  • The two minor comments (abstract and discussion) have been properly addressed in our revised paper.c